# A System Model of Post-Migration Risk Factors Affecting the Mental Health of Unaccompanied Minor Refugees in Austria—A Multi-Step Modeling Process Involving Expert Knowledge from Science and Practice

**DOI:** 10.3390/ijerph17145058

**Published:** 2020-07-14

**Authors:** Nicole Hynek, Arleta Franczukowska, Lydia Rössl, Günther Schreder, Anna Faustmann, Eva Krczal, Isabella Skrivanek, Isolde Sommer, Lukas Zenk

**Affiliations:** 1Department for Knowledge and Communication Management, Danube University Krems, 3500 Krems, Austria; guenther.schreder@donau-uni.ac.at (G.S.); Lukas.Zenk@donau-uni.ac.at (L.Z.); 2Department for Economics and Health, Danube University Krems, 3500 Krems, Austria; Arleta.Franczukowska@donau-uni.ac.at (A.F.); Eva.Krczal@donau-uni.ac.at (E.K.); 3Department for Migration and Globalisation, Danube University Krems, 3500 Krems, Austria; lydia.roessl@donau-uni.ac.at (L.R.); Anna.Faustmann@donau-uni.ac.at (A.F.); Isabella.Skrivanek@donau-uni.ac.at (I.S.); 4Department for Evidence-Based Medicine und Evaluation, Danube University Krems, 3500 Krems, Austria; Isolde.Sommer@donau-uni.ac.at

**Keywords:** system models, expert knowledge, fuzzy-logic cognitive mapping, unaccompanied minor refugees, mental health, post-migration risk factors

## Abstract

Various studies have indicated that unaccompanied minor refugees (UMRs) have a higher risk of suffering from mental health problems than do accompanied minor refugees and general population norm. However, only a few studies provide data on UMRs regarding post-migration risk factors, their interrelations, and their influence on mental health. In this study, system models of post-migration risk factors for mental health and their interactions were developed in the case of Austria. In three consecutive interactive workshops with scientists and practitioners, fuzzy-logic cognitive mapping techniques were used to integrate the experts’ knowledge. The resulting final system model consists of 11 risk factors (e.g., social contacts in the host country, housing situation, or professional health care services). The model provides a deeper insight into the complexity of interrelated direct, indirect, and reciprocal relations, as well as self-reinforcing triads. This systemic approach provides a sound basis for further investigations, taking into account the inherent complex multifactorial dependencies in this topic.

## 1. Introduction

During the refugee movements in the years 2014 to 2016, European countries faced a rapidly growing number of refugees and asylum seekers, resulting in a record level of 1.3 million asylum applications in 2015 [1]. Of these, 8277 asylum applications were filed by under-age refugees [2]. Although the number of asylum-seeking applications has been decreasing since 2016, asylum seekers and refugees still represent a persistent issue and an important field of action, especially in terms of the vital matter of ensuring their mental health and well-being.

Refugees may suffer from mental disorders, as they often cope with dire situations and life events that affect their mental health. In particular, numerous studies have highlighted the special vulnerability of refugee children and adolescents to mental health problems and psychiatric disorders [2,3,4,5,6,7,8]. In a systematic review [3] that included 47 studies from 14 European countries and published from 1990 to 2017, covering a sample size of 24,786 refugee or asylum-seeking minors, the estimated point prevalence of diverse psychiatric disorders and mental health problems for children and adolescents was reported as follows: For posttraumatic stress disorder (PTSD), between 19.0% and 52.7%; for depression, between 10.3% and 32.8%; for anxiety disorders, between 8.7% and 31.6%; and for emotional and behavioral problems, between 19.8% and 35.0%. Despite the highly heterogeneous evidence base, it can be assumed that up to one-third of refugee and asylum-seeking children and adolescents suffer from depression, anxiety disorders, or emotional or behavioral problems, and up to one-half could be affected by PTSD [3]. Moreover, various studies showed that unaccompanied minor refugees (UMR) had a higher risk of suffering from mental health problems and psychiatric disorders than did the accompanied minor refugees and general population norm (cf., [3]). Out of this group, female UMRs appear to be exposed to a higher risk for developing mental health problems, PTSD and depression [4]. Several risk factors can influence the point of prevalence of mental health problems in minors. The literature differentiates between factors of pre-migration, e.g., traumatic exposure to poverty, violence, and war; factors of peri-migration, e.g., separation, sexual abuse, and trafficking; and factors of post-migration, e.g., access to education, social support, the asylum application process, discrimination, acculturation, insecure living conditions, and uncertainty about the future [9,10,11,12,13]. The latter factors are those that can be triggered by country-specific policies of the host countries through the creation of conditions that either hinder or facilitate the integration of refugee children and adolescents and, consequently, either reinforce or weaken their mental health. A comparison of approaches to the integration of UMRs established by EU Member States (MS) shows that MS generally give priority to the care of UMRs. Thereby, MS apply similar accommodation arrangements, appoint a representative (e.g., guardian), provide access to education, the labor market, social welfare assistance and health care (including emergency treatment, basic medical care, and in many cases additional specialized medical care and counseling) [5]. Despite these similarities, integration policies and processes generally depend on the country-specific environment and, hence, post-migration risk factors may be more or less subject to social, political, and economic conditions.

One of the main challenges for the integration of UMRs found in the above-mentioned comparison refers to a lack of specialized and trained staff [5]. This highlights the importance of political and practical recommendations for coping with post-migration risk factors, taking into account the specific conditions affecting the situation of UMRs in a foreign country.

Given the complex interplay between post-migration risk factors, mental health status, and country-specific integration conditions, this study focuses on Austria as a country recording a high share of (unaccompanied) minor asylum applicants [1]. However, in the case of Austria, only a few studies provide data on post-migration risk factors and their influence on the mental health of UMRs. Among others, the Institute for Empirical Social Research (IFES) carried out an exploratory study on the living conditions of UMRs in eastern Austria [14]. This study included quantitative and qualitative interviews with 66 UMRs who spoke about their living situations and future expectations. The study provides insights into essential framework conditions such as accommodation, access to education and work, the financial situation and daily routines of UMRs, and their future perspectives, even if the influence on mental health was not specifically surveyed. Other available publications on specific post-migration risk factors are mostly from non-profit-organizations (e.g., Asylkoordination Österreich, Caritas), international institutions (e.g., European Migration Network), and government institutions and ministries (e.g., Ombudsman Board, the Federal Ministry of Education, Science and Research), though they focus mainly on practical and/or legal matters.

Moreover, significant interrelationships between risk factors have not yet been sufficiently investigated. Consequently, the interactions and dynamics of risk factors of this complex domain of research may be inadequately addressed and implemented in the development of strategies or measurements. In the case of UMRs, this could not only raise costs for secondary and tertiary professional care but also hinder their sustainable integration into society. Thus, evaluating the effectiveness of appropriate measures to ensure the mental health of UMRs requires more than the mere aggregation of parts of the interrelated and multidimensional factors contributing to vulnerabilities of UMRs.

In view of the topicality and relevance of the subject and the scarce availability of data at national and international levels, this study aims to identify post-migration factors, their interrelationships, and potential influence on mental health. An interdisciplinary and systemic approach incorporating experts’ knowledge and experiences was applied to obtain different perspectives from both a scientific and a practical point of view. The final goal of this study was to develop a knowledge-based system model that integrates different knowledge and describes post-migration risk factors as a network of interacting factors. In research and practice, knowledge-based system modeling techniques are used to retrieve the knowledge and represent how individuals organize knowledge, link concepts within a knowledge domain, and understand complex problem situations [15,16,17]. Different types of problems can benefit from this approach of incorporating experts’ knowledge, if, for instance, scientific data on a specific topic is limited or if the problem involves many parties and has no clear solution or clearly correct answers and is, therefore, complex [18,19,20].

In this study, we incorporated the approach of system modeling, as described in the section *Materials and Methods*. Based on our final system model developed within three workshops as described in the *Procedure* section, the research results are presented and discussed within the context of the specific legal and policy framework in Austria. The developed system model calls for a more detailed analysis and should currently only provide insights into future needs for action and further investigation as described in the *Limitations* and *Conclusions*.

## 2. Materials and Methods

### 2.1. The Technique of Fuzzy-Logic Cognitive Mapping

In this study, the technique of fuzzy-logic cognitive mapping (FCM), a commonly used form of semi-quantitative system modeling, was applied. It is an established technique for eliciting, capturing, and diagramming structured knowledge on interrelated issues of a knowledge domain held by individuals or groups [21,22]. The result of the process of system modeling is a cognitive map that takes the form of a system and provides a visual representation of a person’s existing understanding on a particular subject. A cognitive map has three characteristics [23]. The first characteristic includes both the direction and nature of causality; the second characteristic reveals the strength of the connections, and the third characteristic reflects a feedback mechanism that captures the effect of a change in a node on another node that, in turn, affects other nodes along the path. In other words, these cognitive maps contain nodes, i.e., concepts that can be linked qualitatively (e.g., low, medium, high) or quantitatively (e.g., between −1 and 1) to other concepts. This connectivity allows researchers to uncover trends by measuring the degree of conceptual agreement in the cognitive maps produced by individuals [15] or by groups jointly defining, expanding, discussing, and collaborating on the concepts and structures of a system [18].

Furthermore, by applying mapping techniques in a group environment, the knowledge of the group can be extended by developing and discussing the emerging system models. That is, collaborative system modeling can act as scaffolding tools to create an environment in which diverse group members can share their knowledge [24]. Thus, by generating a representation of the problem to solve a particular situation, individual contributors can build on each other, and actions or ideas can be taken up or complemented by other group members [25].

### 2.2. Participants

The participants who were invited to the first workshop consisted of five female scientists working in the fields of migration studies and health economics (see A1–A5 in Table 1). All of them have several years of experience in research projects on health-related migration issues at the same university but are partly assigned to different departments. The expert group of practitioners invited to the second workshop consisted of three male and two female practitioners who worked directly with UMRs in social or medical fields (see B1–B5 in Table 1). All of them had several years of experience in health-related migration issues in their various fields of expertise. They were not acquainted with each other and had no official organizational connections. The participants in the third workshop were the same scientists who had taken part in the first workshop, including one additional scientist from the field of evidence-based medicine (see C6 in Table 1).

### 2.3. Procedure

Study phases I–III were conducted to create the final system model of post-migration risk factors in Austria that contains integrated information and assumptions about the relationship between elements in the system of risk factors. The aim was to obtain different perspectives from both a scientific and practical point of view. The following phases were implemented.

#### 2.3.1. Phase I: Co-Developing System Models

In workshop 1 with five scientists and workshop 2 with five practitioners, each group co-developed a system model of risk factors of UMRs’ mental health upon arrival in Austria. The research question was as follows: “Which influencing factors negatively affect the mental health of UMRs, whereby UMR is used to define persons under the age of 18 who came to Austria without care between the years 2003 and 2018?” In the case of unclear points, the facilitators supported the groups; otherwise, they kept away from the task execution and simply observed the interaction of the participants. The co-development of a system model and its relationships were elaborated on in four tasks:Task 1: Individual elicitation of influential factors.

Using moderation cards, each participant freely wrote up to 10 factors associated with UMRs’ mental health that he or she considered essential based on his or her knowledge and experience.

Task 2: Collaborative identification of the main factors for the system model.

After the individual elicitation of up to 10 risk factors, the participants presented those risk factors to the group and jointly clustered the cards regarding the topics that were similar and most relevant. In a collaborative effort, they labeled these clusters on new moderation cards and defined the names of the main factors to indicate more general concepts.

Task 3: Collaborative drawing of relations between factors.

After a detailed explanation of how to create FCM, each expert group built a system model and collaboratively drew directed relations between the identified factors. A positive relationship meant a positive effect, i.e., if the value of factor A increases, then the value of factor B also increases, while if the value of factor A decreases, then the value of factor B decreases. A negative relationship meant a negative effect, i.e., if the value of factor A increases, then the value of factor B decreases, while if the value of factor A decreases, then the value of factor B increases. The participants placed the factors written on cards on a flip chart in an arrangement of their choice and drew the relationships.

Task 4: Collaborative decision-making of the strength of effects.

After sketching the essential directed relations, the groups decided upon the strength of each relation. For the group of practitioners, a distinction was made between strong, medium, and light relations. A strong positive relation was marked, for example, with three plus signs (“+++”), while a slightly negative relationship was marked with a minus sign (“−”). Considering that scientists are accustomed to numerical expressions of correlations, they evaluated the interactions between the factors as decimals ranging from −1 to +1. Figure 1 depicts the two system models developed in workshop 1 and workshop 2.

#### 2.3.2. Phase II: Consensus-Based Evaluation of the Main Risk Factors

Workshop 3 was designed to enable the group of scientists of this study, as a larger group of six people, to integrate the factors and assumptions of the two groups’ system models from workshops 1 and 2. The task was to jointly identify those factors the participants regarded as the main risk factors of this study. Thus, all the factors identified in the scientists’ and practitioners’ system models were evaluated until an agreement was reached. The co-evaluation of the risk factors followed three criteria: (1) The degree of impact a factor has on UMRs’ mental health, i.e., the weighted degree; (2) the impact’s effects on other factors, i.e., the outdegree; and (3) the degree of being influenced by other factors, i.e., the indegree.

#### 2.3.3. Phase III: From Individual Models to a Shared System Model

After defining the main risk factors of the final system model of this study, the researchers of workshop 3 rated the relationships between these factors and the impact of each factor on mental health. An Excel workbook was created, with one sheet providing the introduction and an example of the task and a second sheet providing a matrix of the factors identified in phase II of the study. In the matrix, the six researchers individually rated all relations between all factors from 0 (no influence) to 10 (most substantial influence). Additionally, the main risk factors were rated regarding their impact on mental health between 0 (no impact) and 10 (most substantial impact). In this sense, the scientists’ mental models, i.e., their knowledge, beliefs, and assumptions about the risk factors of UMRs, were individually gathered to calculate a shared system model of post-migration risk factors of UMRs in Austria. This shared system model is the final model of this study and was calculated by aggregating and filtering the data from all scientists. This procedure allows for the identification of the main relational patterns between the factors. The following thresholds were defined to count a relation in the final system model: Two-thirds of the group (i.e., at least four persons) indicated a value higher than 0 (from 0, no influence, to 10, most substantial influence), and the aggregated average score was higher than 5.0. Afterward, the calculated matrix was visualized as a system model, using the freely available software package Visone. Phase III ended with a sense-making workshop; the researchers evaluated the final system model to identify missing factors or correlations and discussed relational patterns by referring to the current literature. The results of this sense-making process are described in the Discussion and Conclusions sections.

## 3. Results

### 3.1. Experts’ System Models on Post-Migration Risk Factors

Figure 2 depicts the system models of the expert groups in workshops 1 and 2 of this study as well as the final system model evaluated in the course of this study. In all three system models, the positive relations between the nodes are represented as green arrows in three categories: Medium influence (5.0–5.9, light green), strong influence (6.0–6.9, green), and powerful influence (>7.0, dark green). Likewise, the red arrows, i.e., the light red, orange, and red ones in the system models of workshops 1 and 2, represent the negative relations between the factors, whereby they were applied only in these two networks. Additionally, the scientists of workshop 3 rated the main risk factors of the final system model in terms of their impact on mental health. In Figure 2, these ratings are visualized as blue nodes in three categories: Medium impact (5.0–5.9, light blue), strong impact (6.0–6.9, blue), and powerful impact (>7.0, dark blue).

The derivation of the main risk factors of the final system model during the third workshop can be found in Table 2. Congruent factors were identified and maintained, similar factors were summarized, and rarely named factors were removed. As can be seen in Table 2, some factors were not included in the final system model, even though they were named in both workshops.

As in the case of the factor *future perspectives*, the scientists argued that this factor was already covered by the remaining risk factors, such as *income security* and *residence security*. They agreed to exclude the factors *criminal conduct* and *prior information about the dangers of flight and circumstances in the EU* because these had been classified as a mix of post- and pre-migration factors.

Furthermore, the factor *discrimination* was not included in the final system model because it has been considered the possible output of the determinant *political and social climate of the host country*. The factor *contact with family and family remittances*, which includes, among others, financial family support, was initially subsumed under the factor *income security* and excluded afterward. The scientists of workshop 3 considered that the host country cannot influence this factor, and it was ambiguous as the factor of *family reunification*, which was also subsequently excluded. The two factors were excluded for the following reasons: First, the two factors may exert both a positive impact and a negative impact on the mental health of UMRs. For example, the prospect of family reunification can be both a positive future perspective and a source of pressure, whichever is more likely is related to the status of the UMR and the expectations of the family. The same applies to *financial support* for or from the family in the country of origin. Second, the factors have a strong linkage to the circumstances and conditions of the country of origin. Hence, it could not be clearly distinguished as a post-migration risk factor.

Table 3 lists the factors of the final system model depicted in Figure 2 and ranks them by the average score, as determined by the researchers, of the indicated impact on mental health. The network metric indegree calculates the incoming arrows (the factors that influence this specific factor), while outdegree calculates the outgoing arrows (the factors that this specific factor influences). The absolute number measures the amount of incoming or outgoing arrows, while weight measures the percentage of the weighted relations, i.e., the strength of the influence. In this context, a factor with a high absolute indegree is influenced by many other factors, while a high weighted indegree indicates that the factor is strongly influenced by other factors with a high impact, regardless of the amount of the other factors. For instance, the factor social contacts is influenced by many other factors (absolute indegree of 6) that are also highly weighted (weighted impact of 21%). The outdegree signifies the extent to which a specific factor influences other factors, e.g., health care influences a high number of other factors (absolute outdegree of 5) that are also highly weighted (weighted impact of 17%).

### 3.2. Main Risk Factors of the Final System Model

The resulting final system model, as depicted at the bottom of Figure 2, consists of *11 main risk factors*, listed and described below in descending order of their impact on the mental health of UMRs. For better orientation, the impact on UMRs’ mental health on the respective factor is listed in parentheses:The factor *social contacts and relationships in the host country* (0.78) includes UMRs’ formal and informal relationships in the host country, such as contacts and relationships with social workers, caregivers, and friends.*Housing situation* (0.77) refers to the living arrangements and conditions of UMRs, including the size and type of accommodation (living with a foster family or in a residential group, a residential home, supervised accommodation), and the number of roommates and reallocations.The factor *professional health care services* (0.77) involves access to, and the quality of, health care utilization in the host country, including health assessments, public hospital treatment, psychological treatment, and medication.The factor *professional social care services* (0.75) involves access to, and the quality of, social counseling and care involving legal advice and representation, supervising, school enrollment, educational assistance, assistance with administrative procedures, clarification of prospects and the facilitation of family reunification, organization of leisure and recreational activities, and asset management [26].The determinant *daily structure and leisure activities* (0.72) includes access to, and the structuring of, daily and leisure activities (recreation, education, group and individual activities, sports, and household tasks) to ensure a regular daily routine. This factor does not include employment opportunities in the sense of work.*Residence security* (0.70) represents the right of UMRs to reside in the host country; thus, it is related to legal certainty. It depends on the duration of the asylum application process and the obtaining of residence status.*Access to education and training in the host country* (0.67) describes the ability of UMRs to participate, and have equal opportunity, in education and training. This factor plays an important role in dismantling barriers to acquiring educational qualifications and job training and barriers to enhancing sustainable integration into the labor market.*Income security* (0.65) describes the ability of UMRs to meet their basic needs without being afraid of losing their source of income. It depends primarily on social benefits (e.g., needs-based minimum benefit system) and access to education and the labor market.*Sociocultural adaptation* (0.58) is defined as the ability of UMRs to fit into the cultural environment of the host country by adopting its specific cultural elements, such as words, values, behaviors, etc. It refers to the individual competence to handle social interactions and the problems of daily life in a new culture. This narrowed definition was chosen due to the focus, discussions, and results of the workshops. More comprehensively, adaptation is often understood as part of acculturation. Definitions vary; acculturation is a long-term process that encompasses social, systemic, individual/psychological, cultural, and group-dynamic factors [27].The factor *political and social climate in the host country* (0.53) refers to the attitude of politics and society in the host country with regard to refugees—specifically, their degree of openness, acceptance, and readiness to take in refugees. In view of the multidimensionality and multi-faceted nature of this factor, it is supplemented by the inclusion of possible resulting experiences of discrimination by refugees.The factor *German language skills* (0.52) describes the ability of UMRs to use the language (speaking, listening, reading, and writing) of the host country as an essential requirement for effective and proper interpersonal communication.

### 3.3. Description of the Final System Model

The final system model comprehends post-migration risk factors on the individual and societal levels as well as factors combining characteristics of both. However, a strict differentiation of individual and societal factors is not possible. As depicted in Figure 2, at the bottom left of the final system model, most of the factors are on an individual level or lie within the individual’s sphere of influence. Furthermore, they are linked to the general and individual conditions in the country of origin, e.g., *social contacts and relationships in the host country, German language skills*, *daily structure and leisure activities*, and *sociocultural adaptation*.

The factor *social contacts and relationships in the host country*, left in the model, was rated as a factor having a powerful impact on the mental health of UMRs. This factor also showed the highest number of six incoming ties: *Political and social climate in the host country*, *German language skills*, *sociocultural adaptation*, *daily structure and leisure activities*, *professional social care services*, and *access to education and training in the host country*. On the lower left in the model, a triad with reciprocal relations consists of the factors *sociocultural adaptation*, *German language skills*, and *social contacts and relationships in the host country*. A closer look at the factors *sociocultural adaptation* and *social contacts and relationships in the host country* reveals the reciprocal influence and direct as well as indirect levels of impact: While *sociocultural adaptation* describes the ability and individual competence to adjust to new social and cultural settings, the factor *social contacts and relationships in the host country* describes a possible consequence of this competence, but also has an influence on the development and extension and application of this ability (sociocultural adaptation). Social contacts, exchanges and relationships are one possible way of acquiring and applying knowledge and skills about language, culture, and the societal context. Furthermore, *sociocultural adaptation* has a powerful effect on *German language skills* and, therefore, both indirectly and directly influence *social contacts and relationships in the host country*.

The factor German language skills has the highest number of strong reciprocal relationships with other factors. Among these factors are access to education and training in the host country, social contacts and relationships in the host country, and sociocultural adaptation. Only the influence on income security is one-directional. In addition, the factor German language skills is indirectly influenced by political and social climate in the host country via the powerful factor social contacts and relationships in the host country, and itself indirectly influences other factors that have a powerful impact on the mental health of UMRs: Daily structure and leisure activities via access to education and training in the host country, as well as professional health care services and housing situation via income security. The factor daily structure and leisure activities is influenced mainly one-sidedly by other factors, with one exception: It has a medium influence on social contacts and relationships in the host country as part of a reciprocal relation.

The factors *professional social care services* and *residence security* describe societal factors and are characterized by a high outdegree. The factor *political and social climate in the host country* shows several noteworthy characteristics: Although it has only a medium impact on the mental health of UMRs, this factor directly influences factors with a strong and powerful effect and indirectly affects numerous other (very) strong factors. No other factor influences this factor in the model but, through direct—and primarily indirect—relations, it indicates a high degree of impact on factors that can strongly influence the mental health of UMRs.

*Professional social care services* is a factor with a high outdegree (five relations with other factors) and a low indegree; it is influenced only by *political and social climate*. All these relations are one-directional. The factors *residence security* and *income security* have a substantial effect on the mental health of UMRs and are of high relevance due to their influence on factors with a powerful impact. Most notably, *income security* influences three factors that have a powerful impact on the mental health of UMRs: *housing situation*, *daily structures and leisure activities*, and *professional health care services*. Furthermore, it is influenced by *residence security* and *access to education and training in the host country* (by shaping job opportunities) as well as by *German language skills*. *Residence security* exerts a medium influence on the factors *housing situation* and *access to education and training in the host country* and is influenced in a common way by *political and social climate*. Furthermore, this factor has a strong impact on *income security*.

Particularly striking is the frequency of reciprocal relationships as well as the strength and number of indirect and direct relations among the five factors of *social contacts and relationships in the host country*, *access to education and training in the host country*, *daily structure and leisure activities*, *German language skills*, and *sociocultural adaptation*. These factors and their interactions can also be influenced—at least in part—at the individual level. This cluster is characterized by an accumulation of five factors with a high indegree as well as high outdegree and high intensity of the (reciprocal) effect to other nodes. This also means that if one factor were to change, all the other factors would be influenced as well.

## 4. Discussion

The further comparison of the main factors with international and country-specific studies on post-migration risk factors of UMRs points out that while many of the identified factors correspond to the results of international research, the multifactorial impact and relations of factors are seldom addressed. A look at the final system model, including its factors and relationships, reveals that the mental health of UMRs results from a complex interplay between individual and societal-level factors as well as factors combining different characteristics.

The societal factor *professional social care services* has a powerful impact on UMRs’ mental health and various other highly relevant post-migration risk factors. Since UMRs must get by without family backing, support programs connecting them to members of their own and new cultures are recommended to reduce psychological distress [28,29]. In Austria, care for UMRs takes place under a guardianship arrangement, irrespective of the UMRs’ residence status. Until they obtain their residence title, UMRs receive care and social benefits from basic welfare support for foreigners in need of aid and protection and, afterward, from the child and youth welfare system, though provisions can vary from province to province [30,31]. A special challenge is the long waiting periods for the appointment of guardianship, which can lead to delayed diagnosis and treatment of any mental health illness [14]. Furthermore, the basic welfare support legislation in Austria does not define minimum requirements related to qualifications for UMRs’ counselors [26,28]. Considering the societal factor *professional health care services*, which also strongly influences UMRs’ mental health, the present situation is as follows: In most EU countries, health examinations of newly arrived refugee children are provided on either a mandatory basis or a voluntary basis [29]. In Austria, UMRs receive health insurance coverage as part of basic welfare support. Once their residency status has been determined, costs are covered by the statutory health insurance and by the child and youth welfare system. Thus, UMRs get the same access to health care as do Austrian child citizens [30]. However, nothing is known about medical assessments specifically designed to identify UMRs’ special health care needs [28], and there is a general short supply of psychotherapeutic and psychiatric services [28,30].

Due to its highest number of relations to other factors and assumed medium impact on UMRs’ mental health, *access to education and training in the host country* represents a critical factor in the final system model. According to the 2017 data, which, however, do not differentiate between UMRs and accompanied children and adolescents, 18,468 asylum seekers were registered in compulsory general schools and secondary general schools in Austria [31]. The UMRs subject to compulsory schooling generally attend schools as “exceptional pupils or students,” regardless of their residence status. This status allows them to take beginners and tutorial language courses to acquire *German language skills*. However, access to vocational training is granted only to a UMR with a residence title [28]. In 2016/2017, transitional courses were established for young refugees without compulsory schooling at vocational secondary and general higher schools [31]. Education and training play an important role in daily structuring, as shown in our final system model. *Daily structure and leisure activities* have been considered to have a powerful impact on UMRs’ mental health and are influenced by various other risk factors. In Austria, the Basic Welfare Support Agreement stipulates the structuring of daily activities. It covers items such as sports, recreation, individual and group activities, and household tasks [28].

The final system model supposes a powerful relation between the *housing situation* and UMRs’ mental health. For center or campus housing, evidence of the impact of living arrangements on the mental health of UMRs is consistent. Living in a reception center was associated with slightly higher risks of internalizing and/or externalizing problems [32], while minors placed in a reception center for adults had higher levels of psychological distress than did those in a youth reception center [33]. However, differences are no longer significant when one adjusts for the outcome of the asylum application [33]. In one study, living in foster care was associated with higher PTSD symptoms than was living more independently [34]. By contrast, another study found that refugee minors living with a family had significantly less depressive symptoms than did those with other living arrangements [8]. PTSD symptoms were also significantly higher among those in low-support living arrangements [35], while living in unsupported accommodation was associated with psychological ill-health [36]. In Austria, UMRs’ accommodation is provided within the framework of either basic welfare support or child and youth welfare [28]. The low accommodation standards in the area of basic welfare support are criticized compared to other socio-educational institutions running under the child and youth welfare system. In particular, serious differences in accommodation standards among provinces were noted [31].

In the final system model, uncertain application processes and missing *residence security* were associated with poor mental health. Considering this result, international studies have conflicting findings. Yet, ultimately, quicker resolution of asylum claims to reduce insecurity and related distress is recommended [26]. In Austria, UMRs’ entitlement to various kinds of benefits depends on the respective residence status [28]. The final system model presented also illustrates the connection between *residence security* and *income security*, which is related to various other risk factors. On an international scale, one study found a significant association between general hassles such as economic hardship and depression [37,38]. In addition, based on a quantitative survey, Reference [39] found an association between household income and mental health. In Austria, access to employment is granted only to UMRs with a residence title. Generally, to take up employment, UMRs must have reached the age of 15 and completed nine years of compulsory schooling [28]. However, several factors mentioned in the literature are not shown in the system model, e.g., the direct relation of income to social contacts. The model depicts the impact of *access to education and training in the host country* on *income security* but does not include the relationship in the other direction.

The factor *social contacts and relationships in the host country* has been considered a very relevant post-migration risk factor due to its powerful impact on UMRs’ mental health and its numerous direct and indirect connections to other factors, as well as due to the fact that it is part of the previously described triad and cluster. The state of research confirms the influence of social contacts and relationships on mental health. Reference [40] found that the number of friends is related to reduced internalizing behavior problems, though no influence on mental health was found in another study, which also found that being exposed to bullying or marital discord were factors that did not influence mental health [41]. As a result of a longitudinal study of unaccompanied refugees in Norway, Reference [8] stated that the level of daily, general difficulties related to family, friends, and school/work is an independent predictor of depressive symptoms. Further studies found that different daily stressors are associated with higher anxiety, depression, or PTSD symptoms [9,37].

In this regard, the *political and social climate in the host country* has a strong influence on social contacts and relationships. Our assumptions are in line with studies from Sweden and Denmark [13,29] that found an association between the experience of discrimination and lower rates of social acceptance, poorer peer relations, and mental health problems among young refugees. Several international studies confirm the influence of *sociocultural adaptation* on the mental health of UMRs. The study of [42] reported that depression decreases when the level of cultural competence increases. Perceived lower levels of discrimination mediated this effect. The same study, and others, also found an association between acculturation-specific difficulties and depression [37,42]. These findings confirm the stated reciprocal relations between *sociocultural adaptation* and *social contacts and relations in the host country*. The acculturation process should include the development of intercultural competence, describing the ability to effectively communicate and interact with people from various cultures based on one’s intercultural knowledge, skills, and attitudes [43].

*German language skills* are a central prerequisite for broad social and societal participation, which are reflected in the final system model as part of the triad with *social contacts and relationships in the host country* and *sociocultural adaptation*. *Sociocultural adaptation* and *German language skills* are factors with reciprocal relations that are relevant in terms of their direct and indirect influences on various factors in the final model. The connections between the language skills and mental health of UMRs have hardly been researched so far. On an international level, only one study, carried out in Denmark, showed a positive effect of self-assessed Danish spelling competency on internalizing behavior [40,44]. In the final system model, language skills gain great importance because of the many indirect relationships and chains of action associated with them. The factor *German language skills* is not only part of the triad of factors at the individual level but is also linked to numerous factors at the societal level. UMRs attending school have, for the duration of the extraordinary status (a maximum of two years), the opportunity to participate in a language course of 11 h per week [14]. German courses for children that are not compulsory for schooling are often held only once or twice a week, which affects UMRs’ acquisition of language skills and hinders rapid integration processes [31].

As illustrated in the system model by the factor *German language skills*, these multifactorial connections can cause the emergence of vicious circles that follow their own dynamics and can thus further endanger, for example, the mental health, social integration, and well-being of UMRs. For example, a good knowledge of German is essential for building *social contacts and relationships in the host country* and facilitating *sociocultural adaptation*. At the same time, *social contacts and relationships in the host country* are relevant for language training (*German language skills*) and for developing the necessary understanding for *sociocultural adaptation*.

## 5. Limitations

While there is evidence of the higher prevalence of mental health problems and psychiatric disorders among UMRs (cf., [3]), the results regarding the influence of pre- and post-migration factors on UMRs’ mental health are mixed. In this study, we adapted system modeling techniques as an exploratory tool for post-migration risk factors impacting the mental health of unaccompanied minor refugees in Austria. Our system model is intended to contribute to a better understanding of the multi-layered interdependencies between individual and societal factors in refugee-receiving countries, as we showed in the case of Austria.

However, our research is subject to some limitations that must be pointed out. Regarding the connections between post-migration factors, the final system model displays only connections with a value of 5.0 or higher to provide a clearer picture of strong interrelations between the main risk factors of the current study. However, lower weighted connections might be relevant for understanding the complex system of post-migration factors and should be considered when one is planning adequate support structures for UMRs. This involves an effect of *German language skills* on *professional health care services*, of *residence security* on *professional social care services*, and of *housing situation* on *daily structure and leisure activities*, as well as a reciprocal relationship between *access to education and training in the host country* and *sociocultural adaptation*.

Furthermore, some of the factors identified in workshop 1 and workshop 2 were excluded in the further course of the third workshop. For instance, the factor *contact with family and family remittances* or the factor of *family reunification* were not integrated into the final system model, despite their assumed relevance as an important determinant of children’s mental health in workshop 1 and workshop 2 as well as in the literature on family support and family situation [36,42,45]. However, both factors seemed ambiguous to the scientists of workshop 3, as they may exert both a positive and negative impact on the mental health of UMRs, as described in more detail in the results. Moreover, the scientists of workshop 3 considered that these factors could not be clearly classified as a post-migration risk factor. In addition, the factors *sociocultural adaptation* and *social contacts and relationships in the host country* were discussed in the two workshops for the finalization of the model. While these two factors may have a strong influence on resilience and the integration process, their characteristics and strengths are already developing in the country of origin. These examples of factors that have been excluded indicate that a complete separation between pre- and post-migration risk factors cannot ultimately be maintained.

The participants in the scientists’ workshop were selected because of their research focus on migration or health, but additional expertise from the fields of psychotherapy, biopsychosocial health, or the legal field could be involved in future studies in order to further expand the system model. We also could have invited practitioners to the third workshop to integrate the perspectives of scientists and practitioners as the final step of model development. These changes in the composition of the workshop participants also could have led to slightly different results. Moreover, we also could have invited UMRs or their families to one of the three workshops to broaden the perspectives of scientists and practitioners through the voices of UMRs. The process of involving refugees in system modeling, however, requires additional sensitivity regarding the way their knowledge based on experience is drawn out, for example, to avoid emotional stress. In this study, we focused on the group of scientists and decided to expand their knowledge elements with those of practitioners. The inclusion of UMRs’ perspective in a further step of system development could be an outlook for a further study.

Overall, the results of the system model depend on the expertise and knowledge of the workshop participants as well as on group-dynamic factors inherent in the discussion process, and additional expertise and workshops could further enrich the results. An objective, generally valid system model for the risk factors of UMR after migration can hardly be created in three workshops. Knowledge-based system modeling represents an explorative approach that aims to integrate the knowledge of the experts involved in system modeling. Thus, it should be emphasized that the figures and tables presented to illustrate the results could have created a sense of objectivity and generalizability, but this was not our intention. We would like to emphasize that system modeling, as we have done, is an exploratory approach, but it has nevertheless produced interesting results. Therefore, the model must still be checked empirically, and the results must be compared more deeply to the existing literature and empirical findings from similar research so the model can be adopted.

Moreover, gender differences have not been considered in the workshop discussions. In fact, female UMRs appear in small sample sizes or are even not considered in research studies on mental health and associated risk factors. However, research suggests that gender might affect the nature of traumatic events experienced during flight, as well as the challenges faced on arrival in terms of adequate accommodation or sociocultural adaptation [4,8]. The integration of gender aspects would be a further step in refining the model.

Finally, it must be noted that UMRs were defined as “children and adolescents under 18 years of age migrating into another country without a parent or caregiver”. Consequently, the final system model focuses only on UMRs under the age of 18. Since living conditions and received support change when UMRs reach the age of majority, there is a need for a separate model for refugees over 18. In general, UMRs’ transition to adulthood usually requires relocation to accommodation for adults and the loss of guardians and key relationships. Nationwide measures or procedures specifically designed to address refugees’ needs before, during, or after their transition to adulthood are not in place in Austria [28].

## 6. Conclusions

Overall, our final system model of post-migration factors provides a sound basis for further investigations. The consideration of complex multifactorial dependencies illustrates, on the one hand, the limitations of this model. On the other hand, it indicates the benefit of this approach in terms of generating further questions and identifying research needs. The method this paper describes can be used in similar settings in which specialized knowledge from different fields of expertise is needed to explain and understand complex interrelations. By showing the relationships between factors through the final system model, it is possible to put diverging results into a concrete context, to show further influencing factors, and, thus, to uncover starting points for further questions and research needs. While in a standard focus group setting, researchers may develop a system model based on the experts’ views, a collaborative approach toward fuzzy-logic cognitive mapping allows participants to establish their model in the meeting, reflect their current beliefs and assumptions, and run scenarios to evaluate the completeness of their prior beliefs and knowledge [21]. Applying the FCM can also be fruitful at a later implementation stage of the model, as inter-professional collaboration and cooperation among different institutions is required to enhance UMRs’ mental health. Further in-depth analysis may shed some light on the relevance of the conjunction of the systemic framework and individual characteristics and resources in relation to pre-, peri-, and post-migration factors, e.g., personal and family background, socio-economic conditions in the country of origin, and migration experiences.

Nevertheless, the system model shows interdependencies that, so far, empirical research has omitted. As next steps, we encourage researchers to empirically test the relationships described in the system model of post-migration factors. For example, an analysis of the triad between *social contacts and relationships in the host country*, *German language skills*, and *sociocultural adaptation* should reveal valuable insights into the reinforcing effect of these factors on UMRs’ mental health. As discussed previously, *sociocultural adaptation* contains characteristics of both pre- and post-migration factors. In the model, this factor is embedded in reciprocal relationships and exerts strong direct and indirect effects on other factors. Therefore, we propose exploring the post-migration determinants of sociocultural adaptation skills, i.e., to explore those dimensions of social adaptation skills that may develop in the host country. Further, we recommend investigating the relationships between the factors influencing *daily structure and leisure activities*, which has a strong effect on UMRs’ mental health in our model. To the best of our knowledge, empirical evidence of the importance of this factor, as well as its interplay with other pre- and post-migration factors, is scarce.

At the practice level, the system model can be used as a basic framework for determining preventive strategies as well as strategies to restore or enhance the mental health of UMRs. The model underpins the necessity of a holistic approach to address UMRs’ mental health problems. Providing easy access to psychotherapeutic support, for example, represents an important protective factor. However, implemented as a single measure, it might not be as effective as a comprehensive approach dedicated to integrating systemic factors such as health and social services, and housing, with the strengthening of individual resources such as social contacts and relationships and the structuring of daily and leisure activities. Refining the model would allow for the definition of institutional interfaces between health and social care providers and for the planning of collaborative action to provide seamless and comprehensive care and support. Given this paper’s methodological emphasis, the required in-depth substantive dialogue on this issue will take place in a subsequent publication.

## Figures and Tables

**Figure 1 ijerph-17-05058-f001:**
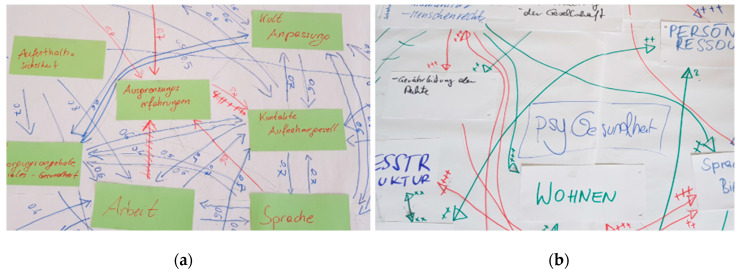
Part of the group system model of post-migration risk factors in Austria co-developed in workshop 1 with scientists (**a**) and workshop 2 with practitioners (**b**).

**Figure 2 ijerph-17-05058-f002:**
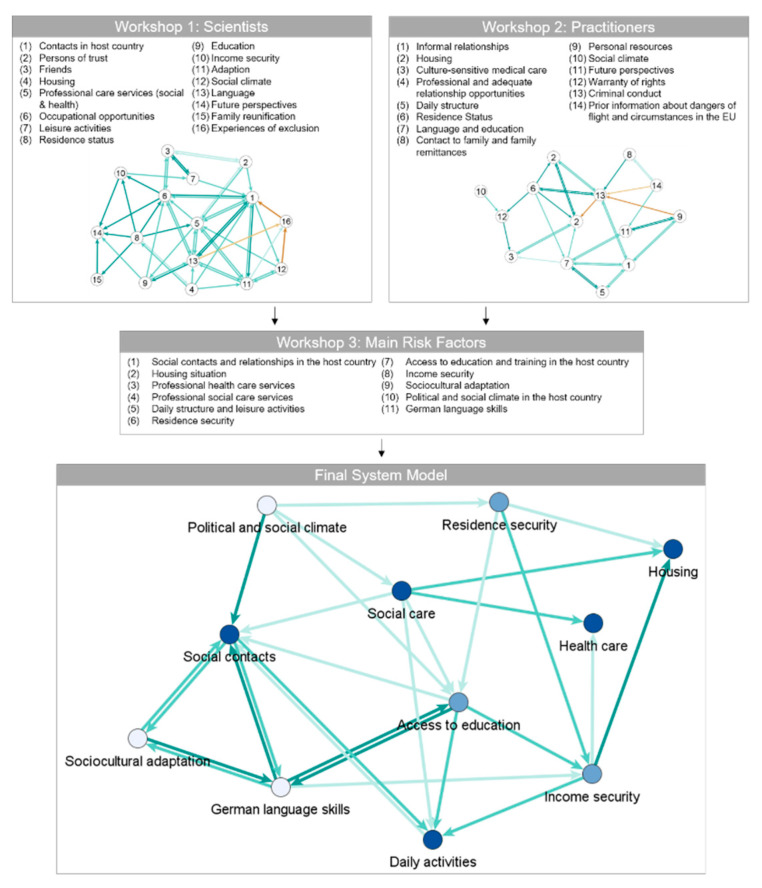
Illustrations depict the system models of post-migration factors developed in workshops 1 and 2. The main factors identified in workshop 3 are illustrated in the center, and the final system model developed throughout the three workshops can be seen below.

**Table 1 ijerph-17-05058-t001:** Participants from workshop 1 to workshop 3.

Participant	Role	Field of Expertise	Years of Expertise
A1	Scientist	Migration studies	10
A2	Scientist	Migration studies	10
A3	Scientist	Migration studies	9
A4	Scientist	Health economics	9
A5	Scientist	Health economics	6
B1	Practitioner	Child and adolescent psychiatry	12
B2	Practitioner	Psychosocial services	2
B3	Practitioner	Social work (youth center)	11
B4	Practitioner	General medicine	25
B5	Practitioner	Education and labor market integration	3
C6	Scientist	Evidence-based medicine	10

A = scientists of workshops 1 and 3, B = practitioners of workshop 2, C = additional scientists of workshop 3.

**Table 2 ijerph-17-05058-t002:** Clustered post-migration risk factors identified in the three workshops.

Main Factors of the Final System Model	Factors of Scientists’ System Model	Factors of Practitioners’ System Model
Social contacts	Contacts in host country	Informal relationships
	Persons of trust	
	Friends	
Housing	Housing	Housing
Health care	Professional care services (social and health) ^1^	Culture-sensitive medical care
Social care	Professional care services (social and health) ^1^	Professional and adequate relationship opportunities
Daily activities	Occupational opportunities	Daily structure
	Leisure activities	
Residence security	Residence status	Residence status
Access to education	Education	Language and education ^2^
Income security	Income security	Contact with family and family remittances
Sociocultural adaptation	Adaptation	Personal resources
Political and social climate	Social climate	Social climate
German language skills	Language	Language and education ^2^
- -	Future perspectives	Future perspectives
- -	Family reunification	- -
- -	Experiences of exclusion	- -
- -	- -	Warranty of rights
- -	- -	Criminal conduct
- -	- -	Prior information about dangers of flight and circumstances in the EU

The factors of the scientists were identified in workshop 1 (N = 5), the factors of the practitioners in workshop 2 (N = 5), and the main factors in workshop 3 (N = 6). The main factors are presented as short names. Professional care services ^1^ (social and health), and language and education ^2^ are single risk factors that have been split into two main risk factors.

**Table 3 ijerph-17-05058-t003:** Network metrics of the final system model.

		Indegree		Outdegree	
Main Risk Factor	Impact	Absolute	Weight in Percent	Absolute	Weight in Percent
Social contacts	0.78	6	21	3	10
Housing	0.77	3	11	0	0
Health care	0.77	2	7	0	0
Social care	0.75	1	3	5	17
Daily activities	0.72	4	13	1	3
Residence security	0.70	1	3	3	9
Access to education	0.67	4	14	4	14
Income security	0.65	3	10	3	11
Sociocultural adaptation	0.58	2	7	2	8
Political and social climate	0.53	0	0	4	13
German language skills	0.52	3	12	4	15

The main risk factors are presented in short names.

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
