# Peer review of "A System Model of Post-Migration Risk Factors Affecting the Mental Health of Unaccompanied Minor Refugees in Austria—A Multi-Step Modeling Process Involving Expert Knowledge from Science and Practice"

_ijerph, 2020, doi:10.3390/ijerph17145058_

Round 1

Reviewer 1 Report

This is an interesting study, particularly in its use of novel methods (fuzzy cognitive mapping) to identify risks/vulnerabilities of mental health in unaccompanied refugee minors.  I appreciated the detailed explanations of the methods and the thorough discussion of benefits and limitations of this methodology.  This paper adds to the literature by describing expert opinions of risks/vulnerabilities for mental health of URMs in a novel way.  The study is well-written and its conclusions are sound.

In general, my most significant feedback is to review the final portion of the results and the discussion - the results section contained some commentary that would be better placed in the discussion.  Otherwise, I have only minor edits.

Specific comments:

Results: Lines 213-215 appear to be copied and pasted in error

Table 2 - inconsistent capitalization of words (Prior Information); please standardize

Results page 10 - earlier in the results, "political and social climate in the host country" was clarified to include the experience of discrimination and also discussed further later on; I recommend adding this information here as well to clarify the scope of this definition.

Results page 11 - some of this information would be better suited to the discussion section of the manuscript.  Any information reflecting on the results or attempting to contextualize the results or interpret them should go in the discussion rather than results section.

The limitations section is well-done, but I would recommend adding a line or two about the fact that the expert opinions upon which the result was based do not include the voices of UMRs or their families.

Author Response

Dear Reviewer 1:

Sincerely,

Nicole Hynek

Reviewer 2 Report

As far as this article is written in English, which means, to broader audiences than those limited to the German-speaking areas, I think the authors should integrate this work into a larger transnational debate. It would be ideal to use the Austrian experience as a trump to larger issues of the field.

While the authors identify the gender of the participants in the workshop - something which might be important - I think it would have been great if they had come up with data about the gender of the migrants that have filled the questionaries. I am sure that gender differences are translated into different trajectories and how migrants react to past, present, and future experiences. Their relationship to their country of origin, the nature of the traumas, and the ability to adapt to the host country is very much shaped by gender identities, sexual orientation, as well as religious affiliation, ethnicity, and so on.  Shedding light over them, I think, would make this article very strong. 

Another element I think the authors of this article should take into consideration is to better clarify some of their statements. For example, in lines 238-239 they state: "The participants consider that the host country cannot influence this factor." In lines 498-499, the authors state again: "The factor contact to family and family remittances was not integrated into the final system model 498 despite its relevance, as highlighted in our workshops and in the literature on family support and 499 family situation, as an important determinant of children's mental health." However, stating is not enough. Explaining "why" they have come up to this conclusion, why the have included or excluded a variable is critical to fully understand the authors' methodology and the way they use the theoretical framework. 

It is unclear what is the exact difference between social contacts and relationships in the host country and sociocultural adaptation. How these two factors do interact with each other? How the social and the cultural do interact with each other and define the levels of adaptation. I think a bit of more clarity helps to understand the analysis of the triad and how its points are interconnected to each other. 

Author Response

Dear Reviewer 2:

Sincerely,

Nicole Hynek
